# Non-Coding RNAs in Rheumatoid Arthritis: Implications for Biomarker Discovery

**DOI:** 10.3390/ncrna8030035

**Published:** 2022-05-25

**Authors:** Julio Enrique Castañeda-Delgado, Noé Macias-Segura, Cesar Ramos-Remus

**Affiliations:** 1Unidad de Investigación Biomédica de Zacatecas, Instituto Mexicano del Seguro Social, Zacatecas 98000, Mexico; 2Cátedras CONACYT, Consejo Nacional de Ciencia y Tecnología, Cd. México 03940, Mexico; 3Departamento de Inmunología, Facultad de Medicina, Universidad Autónoma de Nuevo León, Monterrey 64460, Mexico; noe.maciass@uanl.edu.mx; 4Instituto de Ciencias Biologicas, Universidad Autónoma de Guadalajara, Zapopan 45129, Mexico

**Keywords:** LncRNA, miRNA, biomarker, rheumatic disease

## Abstract

Recent advances in gene expression analysis techniques and increased access to technologies such as microarrays, qPCR arrays, and next-generation sequencing, in the last decade, have led to increased awareness of the complexity of the inflammatory responses that lead to pathology. This finding is also the case for rheumatic diseases, importantly and specifically, rheumatoid arthritis (RA). The coincidence in major genetic and epigenetic regulatory events leading to RA’s inflammatory state is now well-recognized. Research groups have characterized the gene expression profile of early RA patients and identified a group of miRNAs that is particularly abundant in the early stages of the disease and miRNAs associated with treatment responses. In this perspective, we summarize the current state of RNA-based biomarker discovery and the context of technology adoption/implementation due to the COVID-19 pandemic. These advances have great potential for clinical application and could provide preclinical disease detection, follow-up, treatment targets, and biomarkers for treatment response monitoring.

## 1. Novel Gene Expression Analysis Technologies for Biomarkers Discovery

Technologies that analyze gene expression, from qPCR machines to single-cell sequencing for single-cell transcriptomics, have been restricted mostly to research facilities [1]. The translation of these tools to clinical settings has been slow, perhaps due to the few tests being developed that have clinical value (e.g., detecting RNA or DNA from viruses such as HIV and HCV).

Recently, these tools have become available in many centers worldwide due to the COVID-19 pandemic, which required these centers to provide dynamic protocols for RNA detection and/or sequencing [2]. These resources could be used to further significant progress in areas such as inflammatory rheumatic diseases [3]. In the few pages ahead, we highlight the possible application of these technologies in the rheumatology setting using potential RNA-based biomarkers with special emphasis on RA.

## 2. Gene Expression Profiling of Rheumatic Diseases: Focus on RA

Rheumatoid arthritis is a complex disease involving several metabolic pathways, gene polymorphisms, and epigenetic alterations [4]. In this regard, identifying its gene expression profiles is a meaningful strategy to better understand the mechanisms of pathogenicity, disease course and progression, assess therapeutic response, and even treatment selection. Transcriptional profiling is a powerful technology capable of providing an integrative view of pathophysiological pathways. It evaluates thousands of molecules in one experiment [5]. Microarrays (MA) based on high-throughput platforms are among the most important tools for discovering new genetic, transcriptomic, or epigenetic disease alterations and identifying key future biomarkers for diagnosis, prognosis and response to therapy [6,7]. Microarray technology has been applied to compare RA patients’ gene expression profiles in a specific disease stage to healthy controls, other diseases, or another disease status of the same disease [8]. Molecular patterns identified by microarrays can be so specific that they differentiate particularities among patients with the same disease.

Transcriptomic assays supply vast information, which must be simplified to provide some context in systems biology. Many studies in the field use multiple strategies to simplify the metadata. One strategy to simplify gene expression data involves assessing changes in molecular patterns in a specific cell population. For example, dendritic cells (DCs) are important immune cells associated with the genesis of some autoimmune diseases. There is a specific transcriptional signature in DCs identified in three different autoimmune diseases, namely RA, systemic lupus erythematosus, and type 1 diabetes [9,10]. This transcriptomic profile in DCs is associated with two essential characteristics, the heterogeneity of the autoimmune disease and the disease activity level. These immune gene expression profiles provide essential clues to molecular pathways that can serve as determinants for therapy prognosis and disease course. These gene expression profiles can help to identify new molecular targets and treatment development.

We previously reported a specific transcriptional signature in peripheral blood cells associated with first-degree relatives negative to anti-cyclic citrullinated peptide antibodies (ACPA) that differentiate relatives positive to ACPA, and a specific transcriptional signature for treatment-naïve newly diagnosed RA patients [4]. Identifying these gene expression profiles brings new molecular pathways associated with disease course and diagnostic/prognostic candidate biomarkers for RA. One example is identifying the peripheral blood cell biomarkers SUMO1 (Small ubiquitin-like modifier 1) and POU2AF1 (POU domain class 2-associating Factor 1), which have decreased expression in patients with RA compared to first-degree relatives and negatively correlate with disease progression and ACPA levels [11]. Relatives with mild to low levels of these biomarkers can help identify patients with recent-onset RA in the “window of opportunity”, when treatment could have a beneficial effect on disease progression, even leading more effectively to clinical remission of RA [12].

Several studies have already focused on transcriptomics in RA, and have indeed provided plentiful data, some reporting important cell signatures associated with treatment response [6], the participation of T cells in early RA, ACPA [13], molecular differences between first-degree relatives with or without ACPA [4], and many others with substantial information in RA (Table 1). However, it is not currently practical to evaluate the gene expression profile for every patient with RA for diagnosis or to determine treatment failure or success. However, molecular markers obtained by different transcriptomic studies can help identify successful diagnostic biomarkers in clinical settings.

## 3. The Identification of mRNA, LncRNAs, and miRNAs as Biomarkers in RA

As mentioned earlier, technological advances in biomedicine, functional genomics, and systems biology have resulted in the identification of a growing number of biomarkers with potential use in clinical practice. They can help identify patients with early RA, identify treatment failure, differentiate other diseases, and evaluate disease course. Some of these biomarkers come from omics assays, which evaluate thousands of molecules, working with metadata and summarizing information to provide specific differentially expressed molecules as new biomarkers for RA (Table 2). These still require extensive validation in a real-world clinical setting to demonstrate their efficacy.

## 4. Implications and Perspective of Clinical Applicability in RA

Autoimmune diseases are heterogeneous, and RA is pleomorphic in its clinical presentation, disease course, and response to treatment. In this context, identifying RNA biomarkers among patients in different disease stages provides essential information about pathophysiological pathways, functional genetics, and molecular markers for diagnosis, prognosis, and therapy individualization. Due to the advent of RNA-based detection of SARS-CoV-2, the necessary infrastructure needed to detect these RNA species by RT-qPCR is now a reality in most places [21,22].

Ethical, social, and legal implications also need to be considered for the broad adoption of these technologies, such as (1) the anonymity of genetic testing results, (2) data privacy, and (3) the adoption of strict policies and rules regarding information gathering, storage, and access by authorized labs and the biotech industry. These aspects have been widely reviewed in detail elsewhere in order to balance clinical testing usefulness and the prediction of future clinically important conditions against the right to privacy of a patient in the ELSI declaration [23,24]. There has been great advancement in this field given that there are institutions in Mexico, such as the National Institute for Access to Information (INAI, in Spanish), responsible for issuing policies, requirements, storage conditions, and access to personal data that companies, or public entities gather from citizens (genetic data included) and access to non-sensitive government information. In the future, these institutional advances will allow for fast, reliable, and state-of-the-art genetic testing with adequate management of the generated data for the benefit of the patient, investigation, and research.

## Figures and Tables

**Table 1 ncrna-08-00035-t001:** Example of studies using transcriptomics to obtain candidate biomarkers with potential clinical use in RA.

Reference	Technology	Group(s)	Tissue Sample	Key Findings	Clinical Use
[14]	Microarray	RA vs OA	Synovium	Candidate biomarkers used together: IL7R + STAT1 (93.94% Sens; 80.77% Spec)	Diagnostic
[15]	Microarray	RA (early and stablished) vs OA	Synovium	Three candidate biomarkers accordingly to their AUC: GZMA (0.906), PRC1 (0.809) and TTK (0.793)	Diagnostic
[16]	Microarray	RA vs HC	Synovium	Gene modules characterized by the gene expression of CCL5, CCL6, CCL9, CCL10, CCL13, and ADCY2 are potential BM for RA diagnosis	Diagnostic
[4]	Microarray	RA vs FDR	Whole blood	Gene expression profiles associated with RA in high risk relatives, and gene expression of BCL2, SERPINB9, MS4A1, ETS1, EGR1, CX3CL1 and MEF2A are potential BM for RA diagnostic	Diagnostic
[6]	Microarray	RA responders to MTX vs RA nonresponders to MTX	Whole blood	Theoretical model was able to detect ~50% of nonresponders at the expense of a false negative rate of ~20%	Treatment response
[17]	Firefly miRNA detection	Response to tofacitinib treatment	plasma	miRNA signature detection in plasma samples associated with clinical remission or RA flare	Treatment response
[7]	miRNA Microarray	Early RA detection	Whole blood	Identification of early RA cases is possible due to a massive expression of miRNAs in the early phases of disease	Diagnostic
[10]	LncRNAsMicroarray	RA detection	PBMCs	Identification of the transcriptional patterns of expression associated with disease. Among these LncRNAs ENST00000456270 and NR_002838 are promising	Diagnostic

*RA* rheumatoid arthritis, *HC* healthy controls, *OA* osteoarthritis, *BM* biomarkers, *FDR* first degree relatives, *MTX* methotrexate, *AUC* area under curve.

**Table 2 ncrna-08-00035-t002:** Examples of biomarkers in literature with potential clinical use in RA.

Biomarker	AUC	*P*	% Sensitivity	% Specificity	Reference
PCNT	0.742	<0.0001	71.20%	68.60%	[18]
AFF2	0.709	0.0007	50.90%	88.60%	
SIAE	0.713	0.0006	54.20%	82.90%	
RSAD2	0.75	0.044	75.00%	100.00%	[19,20]
LY6E	0.69	0.0581	50.00%	100.00%	
IFI6	0.71	0.0832	62.50%	100.00%	[20]
	0.82	0.005	70.00%	94.74%	[4]
WIF1	0.92	0.001	87.50%	92.86%	[4]
MXA	0.81	0.005	80.00%	80.00%	
SOSTDC1	0.93	<0.001	87.50%	92.86%	

Values of the biomarkers where obtained when compared to healthy controls. (AUC) Area under the ROC curve.

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
