# Peer review of "Non-Coding RNAs in Rheumatoid Arthritis: Implications for Biomarker Discovery"

_ncrna, 2022, doi:10.3390/ncrna8030035_

Round 1

Reviewer 1 Report

Dear Author

I am honored to have the opportunity to review the article entitled “Inflammation control and gene expression regulation through non-coding RNAs in rheumatoid arthritis: Implications for biomarker discovery” again.

In response to my previous comments, the authors have appropriately updated to include the ethical and social implications of genetic analysis technology in the section “Implications and perspective of clinical applicability in RA”.

In addition, they appropriately cited the references that form the basis of discussion in the second paragraph.

The value of this manuscript revised by the authors has gotten better.

This manuscript is considered suitable for publication by this journal in its current content and state except for the requirements to be worked out by the editorial team.

Sincerely yours,

Author Response

Reviewer 1 approved the document. 

No further comments. 

Reviewer 2 Report

I have read the revised version of the manuscript. In the contrary to previous version its quality has been improved, nevertheless, many inaccuracies still persist. Specific comments to be corrected are mentioned bellow.

  • In abstract, please specify "...due to the COVID-19 pandemic."
  • Similarly in the first paragraph, explain why the RNA sequencing tools have become suddenly available e.g. "Recently, these tools have become available in many centers wordlwide due to the COVID-19 pandemics, that required these centers to provide dynamic protocols for RNA detection and/or sequencing..."
  • Sentence "Rheumatoid arthritis is a complex disease in which several molecules, metabolic
    pathways, gene polymorphisms, and epigenetic alterations are involved in the disease's
    development and course" contains a lot of abundant words. I suggest reducing it by omiting the overlined words.
  • Similarly, in the following sentences authors repeat themselves, I suggest rehprasing a deleting the second sentence" Microarray technology has been applied to compare gene expression profiles of RA patients at specific disease stage to healthy controls and  other diseases with similar symptoms. The principal aim of these assays is to dive deep into disease knowledge and differentiate RA from healthy subjects or other diseases with similar signs or symptoms."
  • In the following paragraphs, I do not understand the message from the authors. How do I simplify the omics assays data by focusing on one cell population? Didn't the authors meant that the omics data supply us with so much information, that these data needs to be properly processed and interpretted and in the interprettation it may be helpfull to focus on one cell type?? which will definitelly help in interpretting the data compared to omics of the whole tissues with more cells types, however, the data are definitelly not "simplified" as authors state...  "Transcriptomic assays supply much information, which must be simplified to provide some context in systems biology. Many studies in the field perform multiple strategies to simplify the metadata. One strategy to simplify gene expression data is assessing changes in molecular patterns in a specific cell population. For example, dendritic cells (DCs) are important immune cells associated with the genesis of some autoimmune diseases. .."
  • Sentence "Several studies that evaluate transcriptomics provide much information about RA" shall be rephrased "Several studies have already focused on the transcriptomics in RA and indeed provided plentifull of data, some of the reporting..." and connect this with the following sentence
  • in section 4, please rephrase "about disease course pathways" to "about pathophysiological pathways"

Author Response

Reviewer 2

I have read the revised version of the manuscript. In the contrary to previous version its quality has been improved, nevertheless, many inaccuracies still persist. Specific comments to be corrected are mentioned bellow.

  • In abstract, please specify "...due to the COVID-19"

Response: Accepted suggestion, please see highlighted text

  • Similarly in the first paragraph, explain why the RNA sequencing tools have become suddenly available e.g. "Recently, these tools have becomeavailable in many centers wordlwide due to the COVID-19 pandemics, that required these centers to provide dynamic protocols for RNA detection and/or sequencing..."

Response: Accepted suggestion, please see highlighted text

  • Sentence "Rheumatoid arthritis is a complex disease in which several molecules, metabolic
    pathways, gene polymorphisms, and epigenetic alterations are involved in the disease's
    development and course" contains a lot of abundant words. I suggest reducing it by omiting the overlined words.

Response: Accepted suggestion, please see highlighted text

  •  
  • Similarly, in the following sentences authors repeat themselves, I suggest rehprasing a deleting the second sentence" Microarray technology has been applied to compare gene expression profiles of RA patients at specific disease stage to healthy controls and  other diseases with similar symptoms. The principal aim of these assays is to dive deep into disease knowledge and differentiate RA from healthy subjects or other diseases with similar signs or symptoms."

Response: Accepted suggestion, please see highlighted text

  •  
  • In the following paragraphs, I do not understand the message from the authors. How do I simplifythe omics assays data by focusing on one cell population? Didn't the authors meant that the omics data supply us with so much information, that these data needs to be properly processed and interpretted and in the interprettation it may be helpfull to focus on one cell type?? which will definitelly help in interpretting the data compared to omics of the whole tissues with more cells types, however, the data are definitelly not "simplified" as authors state...  "Transcriptomic assays supply much information, which must be simplified to provide some context in systems biology. Many studies in the field perform multiple strategies to simplify the metadata. One strategy to simplify gene expression data is assessing changes in molecular patterns in a specific cell population. For example, dendritic cells (DCs) are important immune cells associated with the genesis of some autoimmune diseases. .."

Response: Regarding this suggestion, we would like to emphasize that the description of several genes associated with particular immune functions or descriptions regarding specific cell populations are provided as examples of how such changes could be associated with physiopathological features of RA. We do not agree to add the term of “simplify” as suggested by the reviewer for 2 main reasons: 1) A microarray or RNA-seq experiment is not meant to simplify a complex biological phenomenon, but to describe in detail the gene expression changes associated with such complex phenomena, And 2) the graphics or figures associated with such articles are complex and should be treated as such to accurately and thoroughly describe the gene expression changes. Therefore, the term simplify, would not be an accurate description of such complex data.

  • Sentence "Several studies that evaluate transcriptomics provide much information about RA" shall be rephrased "Several studies have already focused on the transcriptomics in RA and indeed provided plentifull of data, some of the reporting..." and connect this with the following sentence

Response: Accepted suggestion, please see highlighted text

  •  
  • in section 4, please rephrase "about disease course pathways" to "about pathophysiological pathways"

Response: Accepted suggestion, please see highlighted text

Round 2

Reviewer 2 Report

I have read the revised version of the manuscript. Authors addressed my comnents adequatelly, I agree with their response on simplification. There is just one now typo in the edited version (section 1: "wordlwid" shall be worldwide). I have no other comments.

Author Response

The typo in "worldwide" has been corrected

This manuscript is a resubmission of an earlier submission. The following is a list of the peer review reports and author responses from that submission.

Round 1

Reviewer 1 Report

I have reviewed the article entitled “Inflammation control and gene expression regulation through non-coding RNAs in rheumatoid arthritis: Implications for biomarker discovery”. This article is very interesting from the following points, and a lot of readers will agree with the author's perspective.

Due to the global COVID-19 pandemic, in a situation where whole of society has a strong interest in this RNA-virus countermeasures, it is unique perspective based on this situation and the authors' works about the rheumatic disease.

The control of COVID-19 is the most important issue in the world, and it should be prioritized, and we would feel that it generates the alteration of funding, social, medical, and human resources. However, we believe that many researchers are competing soundly against common threats within their limitations and are making greater efforts to utilize and advance science and technology. As the authors mentioned, it is thought that this pandemic has led to the widespread recognition of genetic analysis technology as an important tool for the virus control, detection and diagnosis, monitoring of their activity level, follow-up, as well as the evaluation of the effect of vaccination. And this kind of tools and systems would have been installed in many facilities at unprecedented speed.

First of all, the control of this virus is a prerequisite, but we need consider the effective use of these resources in the future, as the authors mentioned.

Also, I request the authors to add and edit the following points in the next version.

1) In general, when science and technology advance, sometimes new issue would be possible to be generated. Sometimes, we need consider the possibility to generate of the victims by inappropriate use of them intendedly or incidentally, rather than developing them into the well-being of humankind and the environment. For instance, the nuclear technology in the 20th century generates not only clean energy and nuclear medicine, but also the health hazards of radiation exposure caused by the atomic bomb and disasters. It is thought that the gene technologies are no exception. Please describe the ethical and social institutional progress also as it may be a little.

2) Please indicate the references to be the basis of discussion. In particular, there is little literature cited in the second paragraph, the reader would covet this information, for example, line64-67, line67-69.

Anyway, this article should lead to encourage the scientists in the other research fields also, and should be published from this journal, biomolecules.

Reviewer 2 Report

I have read with interest the paper by Castañeda-Dalgado et al. about inflammation control in rheumatoid arthritis by non-coding RNAs. Unfotunatelly, the title does not fit the content, article is inapropriatelly referrenced, scientifically misleading, too general and not covering the topic proposed in the title. Specific comments can be found bellow.

Major concerns:

1) The Title „Inflammation control and gene expression regulation through non-coding RNAs in rheumatoid arthritis: Implications for biomarker discovery“ is quiet specific, however, in the manuscript, there are no specific regulatory inflammation-related genes described, nor their regulation by ncRNAs. Only information we have is that there are some genes AND some microRNAs AND some lncRNAs that were tested for their potential biomarker role and that there is a need to confirm such a data; however, there is no data synthesis, no description of inter-connecting pathways, no clear conclusion. Thus I do not see anything novel in the provided text as many similar articles, focusing either on ncRNAs in RA, or other biomarkers in RA, or on pathophysiology of RA, have already been published.

2) Article is inapropriatelly refferenced and scientifically missounding. Already the first sentence „For well-known reasons, there is no doubt that 2020 has been a devastating year for most people worldwide“ would better fit to the newspaper headings than to the scientific article. If authors want to highlight that COVID-19 pandemic increased the use of RNA sequencing and RNA diagnostics, they shall provide good evidence for that, and the same to show that „the year 2020 was devastaring“ - not just stating „for well-known reasons“ - if the article is assessed in 20 years, nobody will know these „well-known reasons“. Such a statement shall be supported by high-quality references documenting the increase in mortality, morbidity, psychological consequences of COVID, traveling restrictions etc. etc.

Moreover, there are also other sentences, that are more scientifically sounding, however, they are not referenced at all, eg.

* Page 2: Sentence „ Rheumatoid arthritis is a complex disease in which several molecules, metabolic pathways, gene polymorphisms, epigenetic alterations, etc., are involved in the disease's development and course.“ need to be properly cited.

* Page 2, lines 65-71: Sentences „There is a specific transcriptional signature in DCs identified in 3 different autoimmune diseases such as RA, systemic lupus erythematosus, and type 1 diabetes. That transcriptomic profile in DCs is associated with two essential characteristics, the heterogeneity of the autoimmune disease and the disease activity level. These immune gene expression profiles give essential clues to molecular pathways that can serve as determinants for therapy prognosis, disease evolution, and they can serve to identify new molecular targets and treatment development“ needs to be properly cited.

3) Information provided in the article are incomplete. Eg. Lines 72 – 82: Authors shall specify in which cells/tissues/fluids transriptional signature were tested (in dencritic cells? anywhere else?) and also where SUMO1 and POU2AF1 were measured (in plasma? serum? dendritic cells again?). These shall be corrected. Also sentence on Page 2, line 61: „ Transcriptomic assays give much information, which must be simplified to give some context in systems biology“ is too general – transriptomics assay definitelly provide much information, but there is no need to simplify them! They just need to be properly intepretted so the information may be used not only in system biology but also in clinical practice.

Minor concerns:

1) The use of abbrevations within the article shall be unified. E.g. Page 1, line 42-43: „Rheumatoid Arthritis“ does not need to be written with capitals and also abbreviation shall be introduced at this location and used further on (e.g. already in the next paragraphs, authors again write „RA“ in full).

2) Article would benefit from slight English proof-reading or just from the author´s proof-reading to remove typos (e.g. page 2, line 87- 88 “ However, by this moment, it is not practical to evaluate the gene expression profile for every patient with RA be diagnose or determine the treatment failure or success.“ - I supposed there shall be „to“ insted of „be“)